# Steroid-Induced Ocular Hypertension in a Pediatric Patient with Acute Lymphoblastic Leukemia: A Case Report

**DOI:** 10.3390/children9030440

**Published:** 2022-03-21

**Authors:** Hung-Yin Lai, Ing-Chou Lai, Po-Chiung Fang, Chih-Cheng Hsiao, Yu-Ting Hsiao

**Affiliations:** 1Department of Ophthalmology, Kaohsiung Chang Gung Memorial Hospital and Chang Gung University College of Medicine, Kaohsiung City 83301, Taiwan; u9801064@cmu.edu.tw (H.-Y.L.); fangpc@cgmh.org.tw (P.-C.F.); born2bunique1008@gmail.com (Y.-T.H.); 2Department of Ophthalmology, Chang Gung Memorial Hospital, Chiayi City 61363, Taiwan; 3School of Medicine, Chang Gung University, Taoyuan City 33302, Taiwan; 4Division of Hematology/Oncology, Department of Pediatrics, Kaohsiung Chang Gung Memorial Hospital and Chang Gung University College of Medicine, Kaohsiung City 83301, Taiwan; chihcheng.hsiao@gmail.com

**Keywords:** ocular hypertension, acute lymphoblastic leukemia, steroid

## Abstract

Glucocorticoids play a pivotal role in therapeutic protocols in acute lymphoblastic leukemia (ALL) treatment. Systemic steroids are known to be less likely to elevate the intraocular pressure when compared to topical administration, and reports addressing hypertensive ocular response in the Asian pediatric ALL population are currently limited. We report a case of a nine-year-old girl with acute lymphoblastic leukemia (ALL) who was found to have highly elevated intraocular pressure (IOP) during maintenance treatment when taking oral dexamethasone (6 mg/m^2^/day). Her IOP increased on day 5 after taking dexamethasone, reached a peak on day 7 or 8, and returned back to baseline on day 13 before anti-glaucoma medications were used. Thus, we prescribed IOP-lowering agents for 10 consecutive days starting on the day oral dexamethasone was administered, and observed that not only did the peak levels lower remarkably, but the IOP levels returned to baseline more rapidly as well.

## 1. Introduction

Glucocorticoids were among the first components used in treating pediatric acute lymphoblastic leukemia (ALL) patients, and still have an essential role in the contemporary treatment protocol [1,2]. Current therapeutic protocols include high dosages of glucocorticoids, which are associated with potential risks for osteonecrosis, metabolic syndrome and obesity, cardiovascular impairment, and nervous system toxic effects. Generally, systemic steroids are less likely to increase intraocular pressure (IOP) compared to topical administration [3]. Studies have revealed that elevated intraocular pressure (IOP) is a possible complication of high-dose corticosteroid use in pediatric hematological malignancies [4,5,6]. de Queiroz Mendonca et al. reported that two of twelve pediatric hematological malignancy cases had steroid-induced ocular hypertension with the use of glucocorticoids [4]. Additionally, Yamashita et al. reported elevated IOP in five Japanese children with ALL [5]. However, no significant difference in intraocular pressure was found in adult patients undertaking low-dose systemic corticosteroid therapy for connective tissue disorders and high-dose intravenous methylprednisolone due to multiple sclerosis when compared to control groups [7,8]. Thus, the change in IOP with systemic corticosteroid use is still controversial.

Here, we report on the case of a child with ALL, who had elevated intraocular pressure during the course of maintenance treatment, and we monitored her IOP fluctuation during the course of systemic corticosteroid treatment.

## 2. Case Presentation

A 9-year-old girl diagnosed with B-cell ALL was referred by a pediatric hematologist with a 3 day history of bilateral blurry halo vision and mild intermittent headache during maintenance phase chemotherapy. She was treated according to the Taiwan Pediatric Oncology Group (TPOG)-ALL-2013 protocol. The doses and timing of steroid usage in this protocol were prednisolone 40 mg/m^2^/day for 4 weeks during the induction therapy, and dexamethasone 12 mg/m^2^/day for 5 days every four weeks in the maintenance phase. The total duration of the maintenance therapy was 120 weeks. The symptoms usually occurred 3–4 days after oral dexamethasone (12 mg/m^2^) was taken. On presentation to our ophthalmology outpatient clinic on day 7 of week 36 of the maintenance phase of TPOG-ALL-2013, the patient’s vital signs were stable and she had no focal neurological signs. Upon ophthalmic examination, her visual acuity was 20/20 in both eyes, and the intraocular pressure with non-contact tonometer was 33.4 mmHg in the right eye and 29.4 mmHg in the left. Slit-lamp exam was unremarkable, with bilateral quiet anterior chambers. The fundoscopy exam revealed pink optic discs with clear disc margins, and normal cup-to-disc ratios (0.3 × 0.3) in both eyes. Gonioscopy showed bilateral 360-degree wide open angle, normal trabecular meshwork pigment and flat iris without any peripheral anterior synechiae. The central corneal thickness was symmetric at 0.600 mm between both eyes. Optical coherence tomography (OCT) (Spectralis OCT, Heidelberg, Germany) of the retinal nerve fiber layer also revealed unremarkable results (104 μm/103 μm).

After monitoring her intraocular pressure (IOP) daily with an electrical applanation tonometer (Tono-Pen^®^ XL applanation tonometer; Reichert Technologies, a unit of Ametek, Inc., Depew, NY, USA) throughout the chemotherapy course, we observed that the patient’s intraocular pressure only increased when oral dexamethasone (12 mg/m^2^/day × 5 days) was used. As the patient’s prior ocular examinations were limited, we used her IOP before oral dexamethasone usage as a baseline, measured on day 1 of week 56 of the maintenance phase of TPOG-ALL-2013. Her baseline IOP was 25/27 mmHg (day 1), 36/32 mmHg four days after oral dexamethasone was used (day 5), and 42/37 mmHg one week after steroid usage (day 8). The patient’s elevated intraocular pressure was treated bilaterally with AZARGA^®^ Eye drops (brinzolamide and timolol) twice a day. Two days after the patient was started on anti-glaucoma medication, her IOP returned back to baseline with 24 mmHg in both eyes. The Azaga^®^ eye drops were then discontinued, and IOP measurements showed no rebound when steroids were not taken. We prescribed Azaga^®^ eyedrops twice a day bilaterally from the first day of oral Dexamethasone use to one week after oral Dexamethasone was discontinued since TPOG-ALL-2013 week 60. IOP remained controlled at below 30 mmHg bilaterally, with 24/25 mmHg on day 5 and 26/25 mmHg on day 8 (Figure 1).

Over the course of treatment, the oral Dexamethasone dosage was reduced to 6 mg/m^2^, half of the recommended dose in the TPOG-ALL-2013 in week 68. However, the patient’s IOP still increased, with the peak on day 7 (36/37 mmHg) and decreased back to baseline (22/20 mmHg) on day 13, without using an anti-glaucoma agent (Figure 2). Therefore, Azaga^®^ eyedrops were prescribed for 10 consecutive days starting on the day oral dexamethasone was administered for each cycle. The IOP remained controlled with the peak on day 8 with 25 mmHg in bilateral eyes without headache or blurred vision. Automated visual field testing (Humphrey^®^ Field Analyzer, Carl Zeiss Meditec; Jena, Germany) showed no visual field defects 31 weeks after initial presentation.

## 3. Discussion

This is a case of steroid-induced ocular hypertensive response in a child with acute lymphoblastic leukemia, which was treated with the Taiwan Pediatric Oncology Group (TPOG)-ALL-2013 protocol. We observed that the patient’s IOP elevated remarkably after oral Dexamethasone use during the maintenance chemotherapy. 

Exogenous corticosteroids may increase intraocular pressure, leading to structural and functional damage of the optic nerve. In children, the elevated IOP and glaucomatous damage are more rapid, and more severe in progression [9]. Thus, steroid usage in children should be cautious. Steroid responders can be divided into low, intermediate, and high responders. Becker used the final IOP to grade steroid responders; thus, lower than 21 mmHg was graded as low, 21–30 mmHg as intermediate, and above 30 mmHg as high [10]. On the other hand, Armaly used the elevation of IOP to rank steroid responders, in which low was considered to be below 6 mmHg, intermediate as 6–15 mmHg, and high as above 15 mmHg [11]. In both studies, the high responders account for about 5–6% of the general population. After steroids are discontinued, IOPs usually normalize within 2 to 4 weeks [12]. High steroid responders require more attention as more intensive damage to optic nerve may ensue.

However, the above steroid responder categorizations are confined to adults rather than the pediatric population [13,14]. Although Goldmann applanation tonometry is considered the gold standard for clinical measurement of IOP, the measurement of IOP using this technique is difficult to carry out in young or uncooperative children. Therefore, the Tono-Pen applanation tonometry is more convenient for children for IOP assessment when IOP is normalized. It was shown that if the IOP was greater than 16 mmHg, the Tono-Pen applanation tonometry tended to mildly overestimate IOP [15]. In our case, we used the same Tono-Pen applanation tonometry to measure IOP. Although it may mildly overestimate when the IOP is elevated, it can still alert us when the IOP rises abnormally.

There are many methods of steroid administration, such as via topical, periocular, intravitreal, inhalation, and systemic routes. IOP rises more frequently in topical administration, and is least likely when administered via the systemic route [3]. In adults, systemic corticosteroids have no significant influence on IOP when used in cases of connective tissue disorder and multiple sclerosis [7,8]. However, young ALL and infantile spasm patients were found to have IOP elevation when high doses of corticosteroids were used [4,5,6,16]. Despite using different doses and duration of corticosteroid under different protocols, the maximum IOP still topped 40–50 mmHg [4,5,6]. It seems that the younger population are more vulnerable to steroid administration. Previous studies showed that patients, both young and old, with primary open angle glaucoma, high myopia, diabetic mellitus, connective tissue disease, pigment dispersion syndrome, endogenous hypercortisolism, have increased risk of developing steroid-induced glaucoma [3]. Steroid-induced ocular hypertension is due to increased resistance in aqueous humor outflow. Cells in the trabecular meshwork have glucocorticoid receptors, which may interact with steroids, causing anatomic changes. These alterations can lead to extracellular matrix deposits and result in increased edema and decreased flow in the trabecular meshwork [3,17]. A mouse model of glucocorticoid-induced glaucoma has confirmed that the dexamethasone increased the secretory protein load of extracellular matrix proteins in the endoplasmic reticulum of trabecular meshwork cells and induced endoplasmic reticulum stress, resulting in ocular hypertension. [6,18] In children with ALL, young age is considered a risk factor. It may be related to the immature structure of their trabecular meshwork. Moreover, previous studies reported that the anterior chamber and its cellular and extracellular constituents become mature at the age of eight [19]. The immature trabecular meshwork may be the reason for their greater response to steroid. Sugiyama et al. found that the IOP of children younger than 9 years old was significantly influenced by dexamethasone compared to those older than 10 years old, which may be related to immature trabecular meshwork [6]. In a prospective study of elevated IOP in children with ALL, the significant risk factors were elevated IOP at ALL diagnosis, female gender, treatment with dexamethasone, and a family history of glaucoma [20]. In fact, our case was a young girl treated with dexamethasone.

Thanks to the progress made in risk-classified chemotherapy, many children with ALL have achieved long-term disease-free survival; therefore, much attention has been paid to the quality of life of the survivors after the treatment. Vision is considered one of the major determinants of quality of life. Nevertheless, the benefits of corticosteroid on a patient’s life outweigh other possible adverse effects of ALL treatment protocols. Elevated ocular pressure was mentioned in previous studies [4,5,6], but not all patients had encountered this adverse effect. Previous studies showed ocular hypertension occurred in about 60–70% of ALL patients with ocular problems [20,21]. As for IOP elevation, a five-year cohort study revealed that the IOP was higher on day 8 of the treatment, and was about 16 mmHg [21]. However, our case is a high steroid responder, in which the maximal IOP was 42 mmHg on Day 8 with a 16 mmHg elevation. Furthermore, we used two courses with different doses of dexamethasone (12 mg/m^2^/day and 6 mg/m^2^/day, respectively), and found more IOP elevation with higher doses of dexamethasone. Sugiyama et al. revealed that the IOP elevation was dose-associated, which was compatible with our case [6]. Generally, IOP elevation may be silent due to poor expression in children [5,6]. Monitoring IOP in pediatric population under corticosteroid treatment is important as ignoring elevated IOP may result in irreversible vision loss due to ocular hypertension.

Detailed history taking and ophthalmologic evaluation are suggested before steroid administration, in order to detect the steroid responders as early as possible. It is important to measure IOP regularly because IOP may be elevated during the treatment course, even in patients whose IOP was within normal range at the beginning of steroid administration. Because the IOP elevation may be silent, children with ALL under chemotherapy should receive a thorough ocular evaluation to monitor the intraocular pressure, even if the optic disc morphologies and visual fields are normal.

## 4. Conclusions

In conclusion, ALL patients who undergo chemotherapy should receive regular follow-up on their ocular status from the beginning of the treatment. Although the IOP was elevated for a short time, IOP control with anti-glaucoma medications are required in high steroid responders in order to prevent further optic nerve damage. Structural and functional examinations are also advised, especially in patients who are considered high steroid responders.

## Figures and Tables

**Figure 1 children-09-00440-f001:**
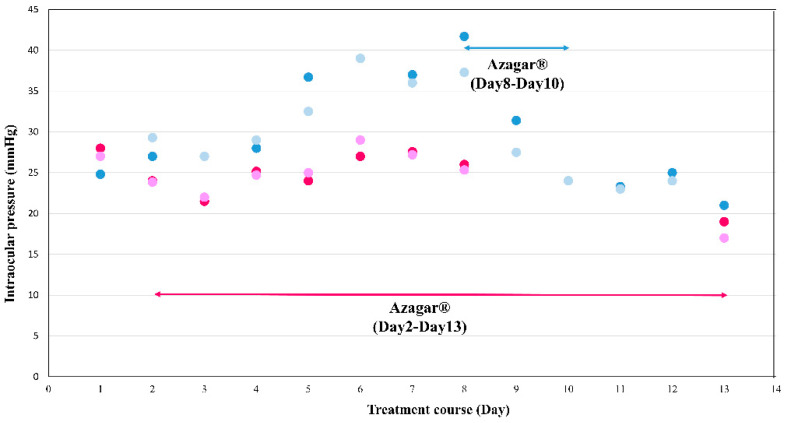
Two courses of intraocular pressure (IOP) fluctuation while using oral dexamethasone (12 mg/m^2^/day). The deep and light blue dots show the IOP change with the use of Azaga^®^ from Day 8–10, while deep and light red dots show the IOP change with the use of Azaga^®^ from Day 2–13. IOP = intraocular pressure. Deep blue and deep red dots represented the IOPs of the right eye, while light blue and light red dots represented the IOPs of the left eye.

**Figure 2 children-09-00440-f002:**
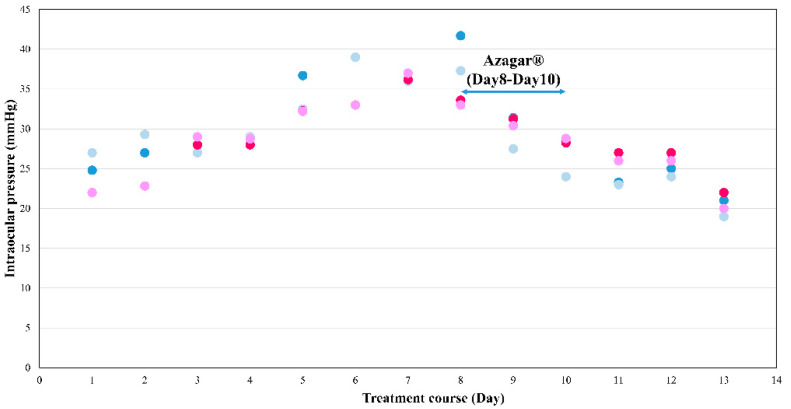
Intraocular pressure fluctuation while using oral dexamethasone under 12 mg/m^2^/day and 6 mg/m^2^/day (12 mg/m^2^/day with blue dots, 6 mg/m^2^/day with red dots). Azaga^®^ was only used at 12 mg/m^2^/day from Day 8–10. Deep blue and deep red dots represent the IOPs of the right eye, while light blue and light red dots represent the IOPs of the left eye.

## Data Availability

The data supporting the findings of this study are private due to the protection of personal data.

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
