# Peer review of "Steroid-Induced Ocular Hypertension in a Pediatric Patient with Acute Lymphoblastic Leukemia: A Case Report"

_children, 2022, doi:10.3390/children9030440_

Round 1
Reviewer 1 Report
Despite significant improvement in ALL treatment outcomes, early and late adverse effects remain a very important problem for all cured patients, especially children's survivors. Corticosteroids are associated with the risk of developing glaucoma and increased intraocular pressure, hence identifying the problem and implementing preventive measures will protect patients against serious complication.
Major revisions:
I propose to extend the discussion to a review of the current literature (2018-2021) on glaucoma in patients with ALL undegoing log-term corticosteroid therapy in the context of patomechanism, diagnostic procedures and the expected long term side effects of the lack of a diagnosed problem. A literature review of this topic will increase the value of the analyzed case report.
Minor revisions:
Please avoid duplicating information - see - Introduction v. 31-33 and Discussion v. 150-152
Reviewer 2 Report
Comments to manuscript children-1597701 Steroid-Induced Ocular Hypertension in a Pediatric Patient with Acute Lymphoblastic Leukemia:A Case Report
This manuscript describes the management of a patient with acute lymphoblastic leukemia (ALL) in whom increased intraocular pressure resulting from systemic administration of dexamethasone was detected. While this same issue has been previously addressed in small group studies, I believe this case is more thoroughly described and still represents a contribution to what has already been reported. Nevertheless, I have some comments that I believe could further highlight the relevance of this work.
- For both the introduction and the discussion, I consider it relevant to integrate other previously published works on the subject (see references), especially because they focus on the pediatric population and the approach and results of each group can be perfectly contrasted. Since all the studies agree on the importance of monitoring the ocular status of pediatric patients with ALL over the course of their chemotherapy, it is appropriate to emphasize this coincidence with a view to unifying diagnostic and treatment criteria.
- In some paragraphs the anti-glaucoma drug is mentioned under the name Azagar, but the correct name is Azarga.
- I suggest that, in Figure 1, the blue arrow be lengthened until day 10, and that it be mentioned that the dark and light points correspond to the change of bilateral IOP.
References:
Yamashita T, Kodama Y, Tanaka M, Yamakiri K, Kawano Y, Sakamoto T. Steroid-induced glaucoma in children with acute lymphoblastic leukemia: a possible complication. J Glaucoma. 2010 Mar;19(3):188-90. doi: 10.1097/IJG.0b013e3181af321d. PMID: 19661822.
de Queiroz Mendonca C, de Souza CP Jr, Martins-Filho PR, Viana SS, Leal BC, Cipolotti R. Steroid-induced ocular hypertensive response in children and adolescents with acute lymphoblastic leukemia and non-Hodgkin lymphoma. Pediatr Blood Cancer. 2014 Nov;61(11):2083-5. doi: 10.1002/pbc.25070. Epub 2014 Aug 17. PMID: 25132159.
Sugiyama M, Terashita Y, Hara K, Cho Y, Iguchi A, Chin S, Manabe A. Corticosteroid-induced glaucoma in pediatric patients with hematological malignancies. Pediatr Blood Cancer. 2019 Dec;66(12):e27977. doi: 10.1002/pbc.27977. Epub 2019 Sep 6. PMID: 31489974.
